# Efficacy of Transarterial Chemoembolization Combined with Molecular Targeted Agents for Unresectable Hepatocellular Carcinoma: A Network Meta-Analysis

**DOI:** 10.3390/cancers14153710

**Published:** 2022-07-29

**Authors:** Zhenzhen Zhang, Yanfang Wu, Tanghui Zheng, Xiaochun Chen, Guobin Chen, Hong Chen, Xinkun Guo, Susu Zheng, Xiaoying Xie, Boheng Zhang

**Affiliations:** 1Department of Hepatic Oncology, Xiamen Clinical Research Center for Cancer Therapy, Zhongshan Hospital (Xiamen), Fudan University, Xiamen 361015, China; zhang.zhenzhen@zsxmhospital.com (Z.Z.); wu.yanfang@zsxmhospital.com (Y.W.); zheng.tanghui@zsxmhospital.com (T.Z.); chen.xiaochun@zsxmhospital.com (X.C.); chen.guobin@zsxmhospital.com (G.C.); chen.hong@zsxmhospital.com (H.C.); guo.xinkun@zsxmhospital.com (X.G.); zheng.susu@zsxmhospital.com (S.Z.); xie.xiaoying@zs-hospital.sh.cn (X.X.); 2Department of Hepatic Oncology, Zhongshan Hospital, Fudan University, Shanghai 200032, China

**Keywords:** transarterial chemoembolization, tyrosine kinase inhibitors, hepatocellular carcinoma, network meta-analysis

## Abstract

**Simple Summary:**

Liver cancer is the second most common cause of cancer-related death, with hepatocellular carcinoma (HCC) being the most prevalent subtype. Transarterial chemoembolization (TACE) in combination with different tyrosine kinase inhibitors (TKIs) has recently been widely used for unresectable HCC (uHCC). However, studies investigating different combinations of agents have shown inconsistent results. Thus, we conducted a network meta-analysis to assess and compare the response of different agents in an uHCC setting. According to our results, TACE plus lenvatinib provides optimal treatment for uHCC, with the highest ranking based on OS, PFS, and DCR rates and the second-best ranking based on ORR rates.

**Abstract:**

Transarterial chemoembolization (TACE) combined with tyrosine kinase inhibitors (TKIs) is the mainstay treatment for unresectable hepatocellular carcinoma (uHCC). However, studies investigating different combinations of agents have shown inconsistent results. Here, we used network meta-analysis (NMA) to compare different agents across 41 studies (36 cohort studies and five RCTs) in 11,540 patients. Multiple RCTs and cohort studies were searched to evaluate TACE combined with different TKIs. Outcomes of interest included overall survival (OS), progression-free survival (PFS), and tumor response. NMA used a random-effects consistency model to pool evidence from direct and indirect comparisons. Hazard ratio (HR) and relative risks (RR) with 95% confidence intervals (CI) were analyzed. Further, heterogeneity and publication bias analyses were performed and agents were ranked. TACE plus lenvatinib provided the maximal OS (Rank probability: 0.7559), PFS (Rank probability: 0.8595), CR (Rank probability: 0.4179), and DCR (Rank probability: 0.3857). TACE plus anlotinib demonstrated the highest PR (*p* = 0.62649) and ORR (*p* = 0.51158). SD was more often associated with TACE plus sorafenib (Rank probability: 0.601685). TACE plus lenvatinib provides optimal treatment for uHCC based on the highest ranking of OS, PFS, and DCR rates. However, given the lack of statistically significant OS benefit, shared decision making should include other TKIs as acceptable alternatives.

## 1. Introduction

Liver cancer ranks fifth in global cancer incidence and second in cancer-related death, with hepatocellular carcinoma (HCC) being the most prevalent subtype [1]. Surgical resection is the major curative treatment for HCC patients. However, approximately 80% of patients have advanced-stage disease at the time of diagnosis and thus have lost their best chance for surgery. There is a 5-year survival rate of only 14.1% [2], leading to a significant burden on the healthcare system.

Treatment options for unresectable HCC (uHCC) are limited. The mainstay of treatment for uHCC includes transplantation, liver locoregional therapy, molecular-targeted therapies, and immunotherapies to prolong survival while preserving or improving quality of life [3]. Although transplantation offers a chance for a cure, it is limited by organ shortages and the need for appropriate patient selection. Liver locoregional therapies are widely used in HCC such as ablation, transarterial chemoembolization (TACE), radioembolization (Y90) and stereotactic body radiation therapy (SBRT). However, these treatments focus primarily on regional disease control. Thus, better locoregional therapy combined with systemic therapies is needed to improve outcomes in patients with uHCC. Tyrosine kinase inhibitors (TKIs) have emerged as the primary treatment methods for uHCC. Sorafenib significantly improved median overall survival (OS) compared to a placebo in a Phase III SHARP trial [4] of more than 10 years. Recently, several other TKIs against uHCC have been approved by the FDA, such as lenvatinib, regarofenib and cabozantinib [5]. In China, apatinib has recently been approved for the treatment of HCC due to its satisfactory efficiency [6]. However, when used alone, the TKIs may have limited efficacy due to the emergence of resistance and side effects. Hence, TACE combined with different TKIs has been widely used recently for uHCC [7,8,9]. On the one hand, TACE can enhance or improve the efficacy and tolerability of TKIs when combined with TKIs. On the other hand, combining anti-angiogenic targeted drugs with TACE has become a promising strategy to decrease post-TACE angiogenesis for better prognoses. However, the role of combination treatment and optimal TKI administration remains controversial [10,11,12,13].

The concept of a combination treatment with different TKIs, instead of monotherapy until progression, has been examined in different randomized clinical trials (RCTs) and cohort studies. Currently, multi-kinase inhibitors (TKIs) such as sorafenib (SOR) [4], lenvatinib (LEN) [14], regorafenib [15], apatinib [16], and anlotinib [17] that potently block the pro-angiogenic pathways are the optimal treatment options for uHCC. Over the years, several trials have been assessed to explore the effects of TKIs combined with TACE for uHCC. However, the clinical efficacy is still controversial, and its potential clinical utility needs to be confirmed. Firstly, trials assessing the combination of sorafenib plus TACE in patients with uHCC have yielded inconsistent results. Three clinical randomized controlled trials (RCTs) (i.e., post-TACE, SPACE, and TACE-2) had unsatisfactory results and failed to demonstrate the clinical benefit of combination therapy over TACE alone [11,18,19]. Several recent systematic reviews and meta-analyses have reported that SOR-TACE was better and safer in treating HCC than TACE alone [8,9,18]. Secondly, two meta-analysis [7,19] demonstrated that apatinib combined with TACE provides better survival benefits for advanced-stage HCC patients than TACE monotherapy. Nevertheless, there is controversy around the efficacy of TACE in combination with apatinib or other TKIs [20,21]. Finally, the combination therapy with LEN-TACE has been heavily investigated recently and the efficacy is promising. A retrospective controlled study in China [22] showed that combination treatment with LEN-TACE may significantly improve PFS and OS benefits over TACE monotherapy, with a manageable side-effect profile for uHCC. Furthermore, a prospective randomized study in China [22] revealed that LEN-TACE was safer, more tolerable and more efficacious compared to SOR-TACE in patients with advanced HCC with PVTT and a large tumor burden.

Taken together, pairwise comparisons between TACE combined with different TKIs was not performed for these RCTs and cohorts, which made it challenging to draw any conclusions regarding which strategy or agent is preferred. Therefore, network meta-analysis is useful to compare different agents across RCTs. In this systematic review and network meta-analysis, we aimed to compare the efficacy of a combination treatment with TACE and different TKIs in patients with uHCC, which included TACE plus sorafenib, TACE plus lenvatinib, TACE plus apatinib, and TACE plus anlotinib.

The study has been registered on PROSPERO (https://www.crd.york.ac.uk/prospero/ (accessed on 1 July 2022)) with the ID CRD42022340934.

## 2. Materials and Methods

### 2.1. Objective

This study aimed to compare the efficacy of TACE combined with different TKIs in patients with uHCC. The reporting of this systematic review follows the Preferred Reporting Items for Systematic Review and Meta-Analyses (PRISMA) statement [23]. Institutional review board approval and informed consent were waived because this was not an individual patient-level meta-analysis.

### 2.2. Eligibility Criteria

The cohort studies and randomized phase 3 clinical trials and were included only if they were published in English. Trials of interest compared TACE with different TKIs in patients with uHCC. Treatment with TKIs included sorafenib, lenvatinib, apatinib, and anlotinib. Studies that included combination therapies other than those mentioned above were also excluded.

### 2.3. Data Sources and Search Strategies

A thorough literature search was conducted for full-text articles published online from database inception through September 2021 from electronic databases such as PubMed, Embase, Web of Science, and the Cochrane Central Register of RCTs and cohort studies. The detailed search strategy is described in Appendix A. The search strategy was designed and conducted by an experienced librarian, with inputs from the study investigators. Two authors identified and reviewed full-text articles that were deemed relevant by screening the titles and abstracts. Disagreements between the two reviewers were resolved by consensus, as needed, with the help of a third investigator. A detailed PRISMA flowchart of the inclusion process is presented in Appendix A.

### 2.4. Data Extraction

Pre-specified data information was extracted from each trial using a structured data form, including baseline characteristics, sample size, tumor burden and interventions used. The independently constructed structured form was used by two investigators for extracting data from the included studies, and disagreements were resolved by referring to a third reviewer. Initial analysis was performed on 25 November 2021. The outcomes of interest included overall survival (OS), progression-free survival (PFS), complete response (CR), partial response (PR), stable disease (SD), progressive disease (PD), objective response rate (ORR), and disease control rate (DCR).

### 2.5. Risk of Bias and Quality Assessment

The methodological quality of cohort studies was assessed using the “star” rating system of the Newcastle–Ottawa Scale (NOS) based on the following three factors: the selection of the research population, the comparability of the study group, and the evaluation of the results. RCTs were graded according to the modified Jadad scale, based on the following items: random sequence generation, concealment of allocation, blinding of participants and personnel, incomplete outcome data, and selective reporting. Disagreements were resolved through mutual discussions by consulting a third investigator when necessary.

### 2.6. Statistical Analysis

We used a Bayesian approach to synthesize data from direct and indirect comparisons of diverse regimens using R 4.1.3 software ((Mathsoft, Cambridge, United States)). OS and PFS were estimated using hazard ratios (HR) with 95% confidence intervals (CI). CR, PR, SD, PD, ORR, DCR, and the odds ratio (OR) with 95% CI were calculated. Two-sided significance was defined as *p* < 0.05. For reported outcomes, fixed- or random-effect models were selected based on model fit criteria (Deviance Information Criteria [DIC]) that penalized increased model complexity [24]. Brooks–Gelman–Rubin diagnostics, traces, and density plots were used to assess convergence. The number of iterations was set to 50,000, and the first 20,000 were used to anneal the algorithm to eliminate the impact of the initial value. Forest plots of the outcome indicators were developed to compare the results. Rank probabilities were calculated to determine the hierarchy of the treatments. The network graph showed an indirect comparative relationship between different interventions and the funnel plots of the outcome indicators tested, and publication bias was described using Stata software (version 15.1, Stata Corporation, College Station, TX, USA). The following treatments were compared: TACE monotherapy, sorafenib monotherapy, TACE plus sorafenib, TACE plus apatinib, TACE plus anlotinib, and TACE plus lenvatinib.

## 3. Results

### 3.1. Study Selection and Baseline Characteristics of Included Studies

Initially, a total of 2952 titles and abstracts were identified using the screening electronic search strategy, of which 384 articles met the eligibility criteria for assessment. Subsequently, 109 review articles, 102 published abstracts or conference proceedings, 128 unrelated topics, and four articles without a study endpoint (PFS or OS) were excluded. Finally, 39 cohort studies and two RCTs were included in the network meta-analysis [10,17,20,21,22,25,26,27,28,29,30,31,32,33,34,35,36,37,38,39,40,41,42,43,44,45,46,47,48,49,50,51,52,53,54,55,56,57,58,59,60]. The PRISMA flow diagram is presented in Appendix A. The baseline characteristics of the included studies are presented in Appendix A. The 41 trials included 11,540 patients. Twenty-one trials used TACE plus sorafenib compared with TACE monotherapy, seven compared it with sorafenib alone and two compared it with TACE plus apatinib. Moreover, seven trials used TACE plus apatinib compared with TACE alone. Two trials compared TACE plus lenvatinib with TACE alone, and one trial compared it with TACE plus sorafenib. Further, one trial used TACE plus anlotinib compared with TACE alone. The ages of the patients in these trials ranged from 18 to 86 years. All patients had BCLC stage B or C and liver function were Child A, or less than B7.

The treatment network is shown in Figure 1, where the thickness of each line in the network plot is proportional to the number of comparisons. Based on the DIC value and I^2^, as described in Appendix A, a random-effects model was applied to the analysis. Sorafenib monotherapy, TACE monotherapy, TACE plus anlotinib, TACE plus apatinib, TACE plus lenvatinib, and TACE plus sorafenib were compared.

### 3.2. Network Meta-Analysis of Clinical Outcomes

#### 3.2.1. Indirect Comparisons of OS

To analyze OS, a network meta-analysis of five different agents was conducted, as described in Table 1. Compared to TACE plus TKIs (apatinib, lenvatinib, or sorafenib), sorafenib or TACE monotherapy had significantly poor OS (HR 2.02, 95% CI 1.26–3.31; HR 2.63, 95% CI 1.23–5.86; HR 1.41, 95% CI 1.01–1.97; HR 2.09, 95% CI 1.50–2.91; HR 2.72, 95% CI 1.37–5.59; and HR 1.46, 95% CI 1.20–1.75, respectively). When pairwise comparisons were performed between the three combined treatment regimens, TACE plus apatinib (HR 0.7, 95% CI 0.49–0.99) demonstrated an OS benefit. Although not statistically significant, TACE plus lenvatinib displayed marginally better OS compared with TACE plus sorafenib (HR 0.535, 95% CI 0.258–1.07) or TACE plus apatinib (HR 0.77, 95% CI 0.35–1.64). Based on the analysis of treatment ranking, TACE plus lenvatinib had the highest likelihood of providing maximal OS, followed by TACE plus apatinib and TACE plus sorafenib (Rank probability: 0.7559, 0.735295 and 0.92354, respectively; Table 2; Figure 2).

#### 3.2.2. Indirect Comparisons of PFS

For PFS analysis, a network meta-analysis of six different agents was conducted, as described in Table 1. Compared to TACE plus lenvatinib, sorafenib monotherapy (HR 3.14, 95% CI 1.45–7.02) or TACE monotherapy (HR 2.99, 95% CI 1.72–5.28) showed poor PFS. Furthermore, in pairwise comparisons between the combined treatment regimens, TACE plus lenvatinib (HR 0.52, 95% CI 0.29–0.92) had significantly better PFS than TACE plus sorafenib. Based on the analysis of treatment ranking, TACE plus lenvatinib had the highest likelihood of providing maximal PFS, followed by TACE plus apatinib and TACE plus sorafenib (Rank probability: 0.8595, 0.35416 and 0.42749, respectively; Table 2; Figure 2).

#### 3.2.3. Indirect Comparisons of CR

To analyze CR, a network meta-analysis of six different agents was conducted, as described in Table 1. Compared to TACE plus TKIs (anlotinib, apatinib, lenvatinib, or sorafenib), sorafenib monotherapy (RR 0.13, 95% CI 0.01–1.34; RR 0.09, 95% CI 0.01–0.67; RR 0.09, 95% CI 0.01–0.73; and RR 0.11, 95% CI 0.01–0.47, respectively) or TACE monotherapy (RR 0.79, 95% CI 0.15–3.96; RR 0.57, 95% CI 0.18–1.64; RR 0.53, 95% CI 0.13–1.9; and RR 0.65, 95% CI 0.3–0.99, respectively) exhibited lower CR rates, although most of them were not statistically significant. When pairwise comparisons were performed between the four combined treatment regimens, no differences were observed in CR rates. However, TACE plus lenvatinib displayed an increased CR trend compared to TACE plus sorafenib. Based on the analysis of treatment ranking, TACE plus lenvatinib had the highest likelihood of providing the maximal CR rate, followed by TACE plus apatinib and TACE plus sorafenib (Rank probability: 0.41794, 0.293095, and 0.434595, respectively; Table 2; Figure 2).

#### 3.2.4. Indirect Comparisons of PR

To analyze PR, a network meta-analysis of six different agents was conducted, as described in Table 1. Compared to TACE plus TKIs (anlotinib, apatinib, lenvatinib, or sorafenib), sorafenib monotherapy (RR 0.25, 95% CI 0.05–1.28, RR 0.41, 95% CI 0.16–1.08, RR 0.38, 95% CI 0.12–1.18 and RR 0.43, 95% CI 0.2–0.91, respectively) or TACE monotherapy (RR 0.34, 95% CI 0.08–1.32, RR 0.55, 95% CI 0.33–0.91, RR 0.51, 95% CI 0.22–1.13, and RR 0.57, 95% CI 0.38–0.83, respectively) exhibited lower PR rates, although most of them were not statistically significant. When pairwise comparisons were performed between the four combined treatment regimens, no differences were observed in the PR rates. Based on the analysis of treatment ranking, TACE plus anlotinib had the highest likelihood of providing the maximal PR rate followed by TACE plus Lenvatinib and TACE plus sorafenib (Rank probability: 0.62649, 0.325115 and 0.3991, respectively; Table 2; Figure 2),

#### 3.2.5. Indirect Comparisons of SD

To analyze SD, a network meta-analysis of six different agents was conducted, as described in Table 1. Sorafenib monotherapy (RR 3.24, 95% CI 1.25–9.86) and TACE monotherapy (RR 3.13, 95% CI 1.28–8.94) displayed higher SD rates than TACE plus anlotinib. Compared to TACE plus apatinib and TACE plus apatinib, TACE plus anlotinib displayed a lower SD rate (RR 0.31, 95% CI 0.11–0.82 and RR 0.28, 95% CI 0.09–0.69, respectively). Based on the analysis of treatment ranking, TACE plus sorafenib had the highest likelihood of providing maximal SD, followed by TACE plus apatinib (Rank probability: 0.601685 and 0.377875, respectively; Table 2; Figure 2).

#### 3.2.6. Indirect Comparisons of PD

To analyze PD, a network meta-analysis of six different agents was conducted, as described in Table 1. Compared to TACE plus TKIs (anlotinib, apatinib, lenvatinib, or sorafenib), sorafenib monotherapy (RR 2.58, 95% CI 0.53–21.46, RR 1.17, 95% CI 0.69–1.92, RR 2.91, 95% CI 1.08–8.57, and RR 1.34, 95% CI 0.91–1.91, respectively) or TACE monotherapy (RR 3.6, 95% CI 0.79–28.85, RR 1.63, 95% CI 1.2–2.24, RR 4.07, 95% CI 1.62–11.39, and RR 1.86, 95% CI 1.48–2.39, respectively) exhibited higher PD trends, although most were not statistically significant. When pairwise comparisons were performed between the four combined treatment regimens, no differences were observed in the PD rates, but TACE plus lenvatinib displayed a lower PD rate than TACE plus sorafenib (RR 0.46, 95% CI 0.17–1.15), TACE plus apatinib (RR 0.4, 95% CI 0.14–1.05), and TACE plus anlotinib (RR 0.89, 95% CI, 0.14–8.55). Based on the analysis of treatment ranking, TACE plus anlotinib had the lowest likelihood of providing minimal PD, followed by TACE plus lenvatinib and TACE plus sorafenib (Rank probability: 0.444345, 0.409235 and 0.60556, respectively; Table 2; Figure 2).

#### 3.2.7. Indirect Comparisons of ORR

To analyze ORR, a network meta-analysis of six different agents was conducted in Table 1. Compared to TACE plus TKIs (anlotinib, apatinib, lenvatinib, or sorafenib), sorafenib monotherapy (RR 0.27, 95% CI 0.06–1.24; RR 0.36, 95% CI 0.14–0.88; RR 0.33, 95% CI 0.11–0.94; and RR 0.36, 95% CI 0.17–0.72, respectively) or TACE monotherapy (RR 0.41, 95% CI 0.11–1.5; RR 0.54, 95% CI 0.33–0.89; RR 0.5, 95% CI 0.23–1.06; and RR 0.55, 95% CI 0.37–0.78, respectively) exhibited lower ORR rates, although some of them were not statistically significant. When pairwise comparisons were performed between the four combined treatment regimens, no differences were observed in the ORR rates; however, TACE plus lenvatinib displayed a higher ORR rate than TACE plus sorafenib (RR 1.1, 95% CI 0.49–2.44). TACE plus anlotinib also displayed a higher ORR rate than TACE plus apatinib (RR1.32, 95% CI 0.33–5.24), TACE plus lenvatinib (RR1.21, 95% CI 0.26–5.4), and TACE plus sorafenib (RR 1.33, 95% CI 0.34–5.00). Based on the analysis of treatment ranking, TACE plus anlotinib had the highest likelihood of providing the maximal ORR, followed by TACE plus lenvatinib and TACE plus sorafenib (Rank probability: 0.51158, 0.29021 and 0.381455, respectively; Table 2; Figure 2).

#### 3.2.8. Indirect Comparisons of DCR

To analyze DCR, a network meta-analysis of six different agents was conducted in Table 1. Compared to TACE plus TKIs (anlotinib, apatinib, lenvatinib, or sorafenib), sorafenib monotherapy (RR 0.93, 95% CI 0.43–2.02; RR 0.71, 95% CI 0.45–1.1; RR 0.71, 95% CI 0.39–1.23; and RR 0.74, 95% CI 0.53–1.03, respectively) or TACE monotherapy (RR 0.88, 95% CI 0.45–1.73; RR 0.67, 95% CI 0.51–0.87; RR 0.68, 95% CI 0.41–1.03; and RR 0.71, 95% CI 0.58–0.85, respectively) exhibited lower DCR rates, although most of them were not statistically significant. When pairwise comparisons were conducted between the four combined treatment regimens, no differences were observed in the DCR rates, but TACE plus lenvatinib displayed a higher DCR rate when compared with TACE plus sorafenib (RR 1.05, 95% CI 0.67–1.71) and TACE plus anlotinib (RR 1.3, 95% CI 0.59–3.02). Based on the analysis of treatment ranking, TACE plus lenvatinib had the highest likelihood of providing maximal DCR, followed by TACE plus sorafenib and TACE plus apatinib (Rank probability: 0.3857, 0.358 and 0.22673, respectively; Table 2; Figure 2).

### 3.3. Results of Quality Assessment, Convergence, Publication Bias, Inconsistency, and Heterogeneity Analyses

Qualitative assessment was performed by assessing various indicators for each study using the NOS for cohort studies or modified JADD scales for RCTs. The results of the quality assessment are summarized in Appendix A. Throughout all analyses, Brooks–Gelman–Rubin was used to confirm preferred model convergence. As described in Appendix A, the potential scale reduction factor was limited to 1, reflecting good convergence in this analysis. The funnel plot of representative studies was nearly symmetrical, suggesting the absence of publication bias (Appendix A). Given the lack of a closed loop in the network graph, inconsistency assessment was not applicable to our study. For all the models, the Gelma–Rubin statistic was greater than 0.05. No inconsistency was observed between indirect and direct evidence of the outcomes of the node-splitting method (Appendix A).

## 4. Discussion

TACE combined with TKIs is currently the main treatment for uHCC, and new combinations are emerging. However, it is difficult to compare these combinations directly. Therefore, in our network meta-analysis, we incorporated direct and indirect evidence to compare the effects of these combinations in patients with uHCC according to their risk categories.

Our network meta-analysis demonstrated that all the combined treatment regimens improved OS, PFS, CR, PR, ORR, and DCR rates compared to sorafenib monotherapy or TACE monotherapy in patients with uHCC. Furthermore, when pairwise comparisons were performed between the four combined treatment regimens, TACE plus lenvatinib displayed improved PFS and better outcomes, with the highest ranking based on OS, PFS, CR, and DCR rates and the second-best ranking based on PR and ORR rates. The results of this study will assist physicians and patients with uHCC in making treatment decisions.

TACE in combination with TKIs has emerged as a potent treatment strategy for improving tumor response and enhancing survival outcomes. Treatment with TACE may lead to the upregulation of vascular endothelial growth factor (VEGF) and fibroblast growth factor (FGF), which stimulate tumor angiogenesis in HCC patients, leading to tumor recurrence or metastasis [61,62]. Furthermore, sequential TACE treatments can result in vascular changes and hepatic dysfunction that ultimately limit the number of TACE treatments that a patient can receive [63]. Therefore, adding TKIs is believed to enhance the effect of TACE by inhibiting tumor cell proliferation and suppressing tumor angiogenesis [64,65]. Furthermore, adding TKIs is also believed to reduce TACE cycles, thus preserving better liver function to enhance patient response and improve tolerance [22,28]. Our results showed that all combined treatment approaches were superior to TACE monotherapy or sorafenib monotherapy, and this finding was consistent with previous research showing that TACE in combination with TKIs was safe and effective [22,66].

Currently, approved TACE-TKI combinations for the treatment of uHCC include TACE plus sorafenib, TACE plus lenvatinib, TACE plus apatinib, and TACE plus anlotinib. Lenvatinib is a multi-TKI with activity against FGF, VEGFR, and MET, and has previously been shown to provide superior OS compared to sorafenib in uHCC [67]. The combination of TACE and lenvatinib has shown promising results based on 1- and 2-year OS rates, and better PFS compared with TACE monotherapy in a retrospective cohort study [22]. Moreover, in a prospective randomized study in China [28], TACE plus lenvatinib showed improved TTP and ORR compared to TACE plus sorafenib. Kudo et al. demonstrated [68] that lenvatinib plus TACE may become the standard therapy for patients who do not benefit from TACE. In our network meta-analysis, TACE plus lenvatinib ranked the highest in terms of OS, PFS, CR, and DCR, and the second was based on PR and ORR. Further, this was consistent with previous studies, Therefore, lenvatinib may be the preferred option for patients with uHCC receiving TACE.

It remains controversial whether treatment with TACE plus apatinib prolongs the survival of patients with uHCC when compared with TACE plus sorafenib. A multicenter retrospective study revealed that TACE–apatinib yielded shorter PFS than TACE-sorafenib and no statistical difference in OS [21], whereas another study demonstrated that TACE plus sorafenib and TACE plus apatinib exhibited comparable prognosis for HCC patients with PVTT [20]. Our network meta-analysis showed that TACE plus apatinib resulted in significantly improved OS compared with TACE plus sorafenib, but the difference was not statistically significant. Furthermore, TACE plus apatinib ranked second highest in our analysis in terms of OS and PFS. Therefore, apatinib may be another option for uHCC patients receiving TACE. In addition, cost-effectiveness is considered another important aspect in selecting the most appropriate regimen for uHCC, and this requires further exploration.

The combination of anlotinib and TACE treatment has the highest ranking for ORR rate, which helps in improving patient compliance and contributes to more opportunities to obtain local treatment, inducing a favorable environment for conversion to surgery. Interestingly, there was a discrepancy between the ranking of ORR and OS/PFS benefits. This discrepancy may have been caused by measurement bias due to the method with which the tumor measurements were taken (in the setting of the subjectivity of RECIST) and when these measurements were made [69]. A prospective study is required to validate these findings.

Our network meta-analysis had some limitations. First, there are some other TKIs available for the treatment of uHCC, such as regorafenib and cabozantinib, but eligible studies evaluating the first-line combination with TACE were not available, which made it impossible to compare the efficacy of TACE plus regorafenib or TACE plus cabozantinib.

However, a recent study assessed the potential benefit of regorafenib in combination treatment with transarterial chemoembolization [66]. However, the study was focused on TACE combined with second-line regorafenib in patients with unresectable advanced HCC and failure of first-line treatment. Our study was focused on the benefits of first-line TKIs in combination with TACE in patients with uHCC. Indeed, this regimen would be a welcomed addition to second-line treatment options in terms of benefits and tolerability. Furthermore, another recent study [70] indicated that the regorafenib-loaded polylactide-co-glycolic acid (PLGA) microspheres prepared for TACE have delivered the promising benefits of limiting proangiogenic responses in liver tumors after TACE and improving the effects of TACE, and could be a novel therapy for TACE. Additionally, in vitro and vivo experiments were performed to verify their effectiveness. Thus, this novel combination strategy with TACE may have promising clinical implications in the future. Second, in the series studies, the majority of participants were Asian, which may have overemphasized the effect of ethnic differences. Finally, this analysis was performed using study-level data rather than individual patient data, which limits the power of the analysis.

## 5. Conclusions

The treatment landscape of advanced HCC has significantly changed over the past few years. The strategies of the TACE plus TKIs are now considered the first-line setting in patients with uHCC. Our study provides the most comprehensive and up-to-date analysis of TACE plus TKIs strategies for uHCC treatment. Lenvatinib plus TACE is a preferred option in patients with uHCC, with sorafenib and apatinib as an additional option when combining them with TACE. Future trials should focus on other potential combinations and the best treatment strategy in patients with uHCC. Data from ongoing head-to-head clinical trials are required to substantiate our findings.

## Figures and Tables

**Figure 1 cancers-14-03710-f001:**
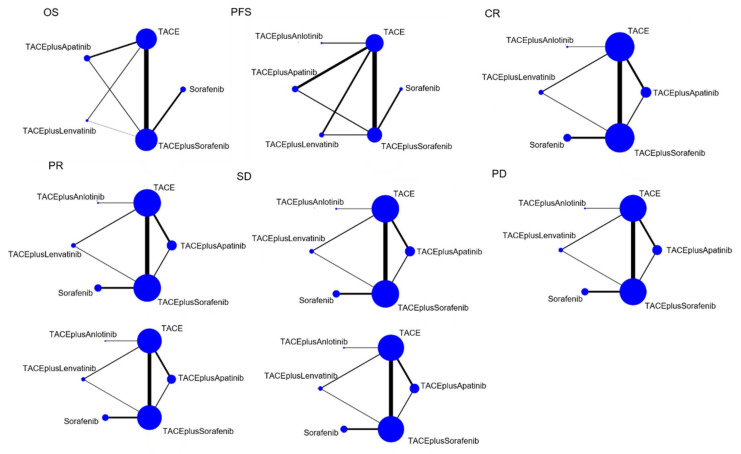
Network graph of the outcomes.

**Figure 2 cancers-14-03710-f002:**
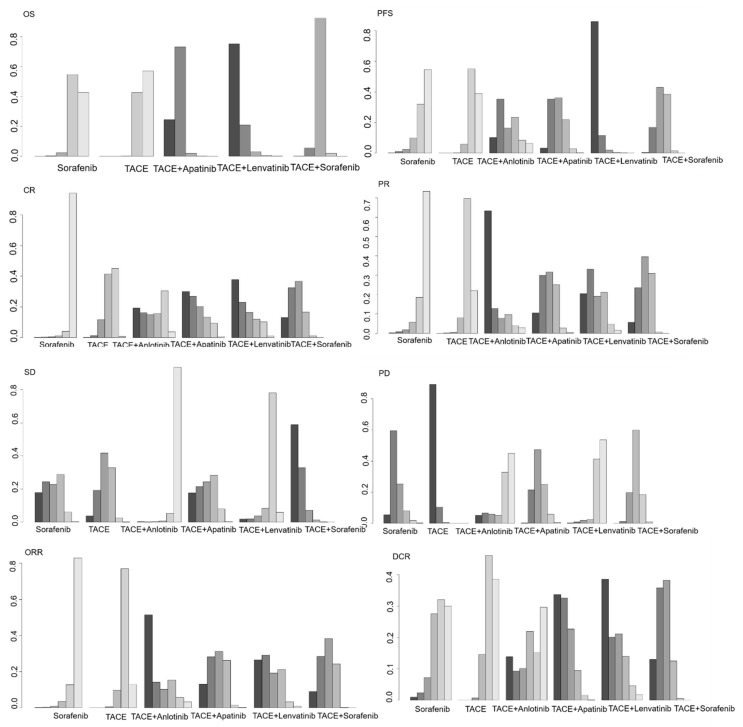
Ranking probabilities of intervention in the studies. The dark-to-light color indicates the rank order. The size of the bar corresponds to the probability of interventions in each treatment.

**Table 1 cancers-14-03710-t001:** League table showing indirect comparisons among TACE plus TKI treatments.

**OS**					
**Sorafenib**					
0.97 (0.66, 1.42)	**TACE**				
**2.02** (**1.26, 3.31**)	**2.09** (**1.5, 2.91**)	**TACE + Apatinib**			
**2.63** (**1.23, 5.86**)	**2.72** (**1.37, 5.59**)	1.3 (0.61, 2.85)	**TACE + Lenvatinib**		
**1.41** (**1.01, 1.97**)	**1.46** (**1.2, 1.75**)	**0.7** (**0.49, 0.99**)	0.54 (0.26, 1.07)	**TACE + Sorafenib**	
**PFS**					
**Sorafenib**					
1.05 (0.57, 1.95)	**TACE**				
1.74 (0.64, 4.87)	1.66 (0.74, 3.74)	**TACE + Anlotinib**			
1.74 (0.9, 3.73)	**1.67** (**1.12, 2.63**)	1 (0.41, 2.56)	**TACE + Apatinib**		
**3.14** (**1.45, 7.02**)	**2.99** (**1.72, 5.28**)	1.8 (0.68, 4.86)	1.79 (0.88, 3.5)	**TACE + Lenvatinib**	
**1.62** (**0.96, 2.84**)	**1.54** (**1.17, 2.08**)	0.93 (0.4, 2.2)	0.93 (0.57, 1.44)	**0.52** (**0.29, 0.92**)	**TACE + Sorafenib**
**CR**					
**Sorafenib**					
**0.17** (**0.02, 0.87**)	**TACE**				
0.13 (0.01, 1.34)	0.79 (0.15, 3.96)	**TACE + Anlotinib**			
**0.09** (**0.01, 0.67**)	0.57 (0.18, 1.64)	0.71 (0.1, 5.04)	**TACE + Apatinib**		
**0.09** (**0.01, 0.73**)	0.53 (0.13, 1.9)	0.67 (0.08, 5.32)	0.94 (0.16, 5.2)	**TACE + Lenvatinib**	
**0.11** (**0.01, 0.47**)	**0.65** (**0.3, 0.99**)	0.81 (0.13, 4.18)	1.12 (0.33, 3.5)	1.21 (0.26, 4.92)	**TACE + Sorafenib**
**PR**					
**Sorafenib**					
0.75 (0.32, 1.78)	**TACE**				
0.25 (0.05, 1.28)	0.34 (0.08, 1.32)	**TACE + Anlotinib**			
0.41 (0.16, 1.08)	**0.55** (**0.33, 0.91**)	1.63 (0.38, 7.29)	**TACE + Apatinib**		
0.38 (0.12, 1.18)	0.51 (0.22, 1.13)	1.52 (0.31, 7.6)	0.93 (0.36, 2.37)	**TACE + Lenvatinib**	
**0.43** (**0.2, 0.91**)	**0.57** (**0.38, 0.83**)	1.71 (0.4, 7.2)	1.05 (0.57, 1.84)	1.13 (0.48, 2.57)	**TACE + Sorafenib**
**SD**					
**Sorafenib**					
1.03 (0.73, 1.51)	**TACE**				
**3.24** (**1.25, 9.86**)	**3.13** (**1.28, 8.94**)	**TACE + Anlotinib**			
1.01 (0.67, 1.67)	0.98 (0.73, 1.38)	**0.31** (**0.11, 0.82**)	**TACE + Apatinib**		
1.46 (0.82, 2.6)	1.42 (0.87, 2.26)	0.45 (0.14, 1.24)	1.44 (0.79, 2.48)	**TACE + Lenvatinib**	
0.89 (0.67, 1.22)	0.87 (0.7, 1.05)	**0.28** (**0.09, 0.69**)	0.88 (0.61, 1.2)	0.61 (0.38, 1)	**TACE + Sorafenib**
**PD**					
**Sorafenib**					
0.72 (0.45, 1.09)	**TACE**				
2.58 (0.53, 21.46)	3.6 (0.79, 28.85)	**TACE + Anlotinib**			
1.17 (0.69, 1.92)	**1.63** (**1.2, 2.24**)	0.45 (0.06, 2.15)	**TACE + Apatinib**		
**2.91** (**1.08, 8.57**)	**4.07** (**1.62, 11.39**)	1.12 (0.12, 7.08)	2.49 (0.95, 7.2)	**TACE + Lenvatinib**	
1.34 (0.91, 1.91)	**1.86** (**1.48, 2.39**)	0.52 (0.06, 2.41)	1.14 (0.81, 1.63)	0.46 (0.17, 1.15)	**TACE + Sorafenib**
**ORR**					
**Sorafenib**					
0.66 (0.29, 1.47)	**TACE**				
0.27 (0.06, 1.24)	0.41 (0.11, 1.5)	**TACE + Anlotinib**			
**0.36** (**0.14, 0.88**)	**0.54** (**0.33, 0.89**)	1.32 (0.33, 5.24)	**TACE + Apatinib**		
**0.33** (**0.11, 0.94**)	0.5 (0.23, 1.06)	1.21 (0.26, 5.4)	0.92 (0.37, 2.24)	**TACE + Lenvatinib**	
**0.36** (**0.17, 0.72**)	**0.55** (**0.37, 0.78**)	1.33 (0.34, 5)	1.01 (0.56, 1.73)	1.1 (0.49, 2.44)	**TACE + Sorafenib**
**DCR**					
**Sorafenib**					
1.05 (0.72, 1.55)	**TACE**				
0.93 (0.43, 2.02)	0.88 (0.45, 1.73)	**TACE + Anlotinib**			
0.71 (0.45, 1.1)	**0.67** (**0.51, 0.87**)	0.76 (0.37, 1.57)	**TACE + Apatinib**		
0.71 (0.39, 1.23)	0.68 (0.41, 1.03)	0.77 (0.33, 1.69)	1.01 (0.58, 1.65)	**TACE + Lenvatinib**	
0.74 (0.53, 1.03)	**0.71** (**0.58, 0.85**)	0.8 (0.4, 1.61)	1.05 (0.78, 1.41)	1.05 (0.67, 1.71)	**TACE + Sorafenib**

**Table 2 cancers-14-03710-t002:** Analysis of treatment ranking probability in patients with uHCC.

Intervention	Rank 1	Rank 2	Rank 3	Rank 4	Rank 5	Rank 6
**OS**						
Sorafenib	0.000165	0.002785	0.02387	0.54582	0.42736	
TACE	0	0.00002	0.00174	0.42741	0.57083	
TACEplusApatinib	0.24607	0.73248	0.019875	0.00156	0.000015	
TACEplusLenvatinib	0.752815	0.20918	0.030835	0.005385	0.001785	
TACEplusSorafenib	0.00095	0.055535	0.92368	0.019825	0.00001	
**PFS**						
Sorafenib	0.001495	0.010155	0.024575	0.098535	0.319985	0.545255
TACE	0	0.00005	0.00137	0.05933	0.550845	0.388405
TACEplusAnlotinib	0.102385	0.352125	0.164635	0.23307	0.08498	0.062805
TACEplusApatinib	0.03351	0.35386	0.36118	0.21973	0.02867	0.00305
TACEplusLenvatinib	0.859055	0.11471	0.01953	0.00542	0.00115	0.000135
TACEplusSorafenib	0.003555	0.1691	0.42871	0.383915	0.01437	0.00035
**CR**						
Sorafenib	0.000045	0.000215	0.00062	0.00209	0.016385	0.980645
TACE	0.000085	0.00653	0.085675	0.40477	0.500935	0.002005
TACEplusAnlotinib	0.189375	0.188725	0.15837	0.14672	0.30269	0.01412
TACEplusApatinib	0.33443	0.293095	0.17729	0.110045	0.08378	0.00136
TACEplusLenvatinib	0.41794	0.24931	0.14345	0.098015	0.08942	0.001865
TACEplusSorafenib	0.058125	0.262125	0.434595	0.23836	0.00679	0.000005
**PR**						
Sorafenib	0.00206	0.00715	0.016925	0.0559	0.18513	0.732835
TACE	0	0.00014	0.004565	0.078665	0.696835	0.219795
TACEplusAnlotinib	0.632085	0.128915	0.07674	0.0959	0.038135	0.028225
TACEplusApatinib	0.105135	0.298485	0.315525	0.25067	0.026925	0.00326
TACEplusLenvatinib	0.204805	0.33027	0.190945	0.210705	0.04741	0.015865
TACEplusSorafenib	0.055915	0.23504	0.3953	0.30816	0.005565	0.00002
**SD**						
Sorafenib	0.17797	0.244155	0.22778	0.28737	0.060625	0.0021
TACE	0.03658	0.191355	0.417	0.32856	0.026175	0.00033
TACEplusAnlotinib	0.002335	0.00165	0.002725	0.005405	0.05199	0.935895
TACEplusApatinib	0.17685	0.21434	0.243985	0.2837	0.078615	0.00251
TACEplusLenvatinib	0.018445	0.02053	0.036525	0.083655	0.781695	0.05915
TACEplusSorafenib	0.58782	0.32797	0.071985	0.01131	0.0009	0.000015
**PD**						
Sorafenib	0.05472	0.59534	0.252775	0.07829	0.017455	0.00142
TACE	0.891825	0.104465	0.00368	0.00003	0	0
TACEplusAnlotinib	0.050625	0.06541	0.05785	0.049785	0.32718	0.44915
TACEplusApatinib	0.001515	0.21449	0.47315	0.24921	0.05787	0.003765
TACEplusLenvatinib	0.00131	0.009085	0.016815	0.022905	0.412635	0.53725
TACEplusSorafenib	0.000005	0.01121	0.19573	0.59978	0.18486	0.008415
**ORR**						
Sorafenib	0.000485	0.001855	0.00743	0.03348	0.127235	0.829515
TACE	0.00001	0.00011	0.00478	0.09716	0.76965	0.12829
TACEplusAnlotinib	0.514745	0.14113	0.101675	0.153605	0.05618	0.032665
TACEplusApatinib	0.129925	0.28231	0.311385	0.26125	0.01362	0.00151
TACEplusLenvatinib	0.264965	0.290715	0.192055	0.212025	0.032225	0.008015
TACEplusSorafenib	0.08987	0.28388	0.382675	0.24248	0.00109	0.000005
**DCR**						
Sorafenib	0.009365	0.02357	0.07199	0.27588	0.320075	0.29912
TACE	0	0.000055	0.006505	0.14567	0.46151	0.38626
TACEplusAnlotinib	0.13843	0.092665	0.101555	0.21938	0.15188	0.29609
TACEplusApatinib	0.336555	0.32534	0.22673	0.09481	0.01573	0.000835
TACEplusLenvatinib	0.385705	0.20037	0.21111	0.139195	0.045935	0.017685
TACEplusSorafenib	0.129945	0.358	0.38211	0.125065	0.00487	0.00001

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
