# Peer review of "Efficacy of Transarterial Chemoembolization Combined with Molecular Targeted Agents for Unresectable Hepatocellular Carcinoma: A Network Meta-Analysis"

_cancers, 2022, doi:10.3390/cancers14153710_

Round 1
Reviewer 1 Report
In this original research paper, the authors conducted a network meta-analysis to evaluate the efficacy of transarterial chemoembolization combined with various tyrosine kinase inhibitors for the treatment of unresectable hepatocellular carcinoma. Detailed information is provided regarding statistical methodology, plots, and analytical results. The reviewer considers that this manuscript provide a topic of interest to the audiences in this field. However, the results mostly confirmed what have been reported in the literature regarding the combination therapy and did not leads to new finding or strategy. In addition, the reviewer found the manuscript challenging to comprehend some of the content and interpret the results. Therefore, the reviewer does not support its publication at cancers.
Reviewer 2 Report
This paper is a very interesting study, a network meta-analysis, to know the results obtained in patients with uHCC with the strategies of the TACE plus TKIs. In addition, it constitutes a starting point for clinical trials in patients with uHCC.
Reviewer 3 Report
The Authors have conducted a very interesting network meta-analysis to compare different TKI in combination with TACE for the treatment of HCC. The analyzed pool of studies is quite large, covering more than 11K patients, and the methods used are sound and well-grounded.
I feel that the manuscript deserves dissemination. I can offer the following comments:
- can the Authors expand on the clinical relevance of their findings?
- statistical analysis section, I feel that some bits of information are not necessary (e.g., the commands used in the package). can the Authors reformulate and move some pieces of information to the Supplementary section?
- quality of figures should be improved for publication
- some English polishing - especially in the simple summary - is required
Reviewer 4 Report
Here the authors present a network meta-analysis (NMA) to compare the efficacy of Transarterial chemoembolization (TACE) combined with tyrosine kinase inhibitors (TKIs). They considered different agents across 41 studies (36 cohort studies and five RCTs) in 11,540 patients. Outcomes of interest included overall survival (OS), progression-free survival (PFS), and tumor response. They found that TACE plus anlotinib demonstrated the highest PR (P =0.62649) and ORR (P =0.51158). SD was more often associated with TACE plus sorafenib (P score: 0.601685). TACE plus lenvatinib provides optimal treatment for uHCC. However, given the lack of statistically significant OS benefit, shared decision-making should include other TKIs as acceptable alternatives and they concluded that their study would suggest that data from ongoing head-to-head clinical trials are required to substantiate these findings.
The study is of interest as a combination treatment based on TACE and TKI is a promising approach. However, some points should be addressed.
-The authors stated in their conclusion that regorafenib was not evaluated because there are no studies assessing combination treatment with TACE and regorafenib. However, they did not include a promising approach described in a study assessing the potential benefit of regorafenib in combination treatment with transarterial chemoembolization. In fact, Regorafenib-loaded polylactide-co-glycolic acid (PLGA) microspheres for improvement of TACE therapeutic effects, which can sustainably deliver regorafenib to limit proangiogenic responses in liver tumors after TACE, has been recently developed (Regorafenib-loaded poly (lactide-co-glycolide) microspheres designed to improve transarterial chemoembolization therapy for hepatocellular carcinoma. Asian J Pharm Sci 2020; 15: 739–751.). In this regard, the authors should discuss the potential added value of regorafenib due to its wider inhibitory effects on several targets as described in a recent comprehensive review (Experience with regorafenib in the treatment of hepatocellular carcinoma. Therap Adv Gastroenterol. 2021 May 28;14:17562848211016959) also describing the immunomodulatory effects of regorafenib.
Round 2
Reviewer 1 Report
The reviewer is convinced that the authors provided sufficient information to address the concerns raised. Therefore, the reviewer supports this manuscript to be considered for publication at Cancers.